# Controlling a simple model of bipedal walking to adapt to a wide range of target step lengths and step frequencies

Sina Mehdizadeh[1*], James Maxwell Donelan[1,2]

1 Department of Biomedical Physiology and Kinesiology, Simon Fraser University, Burnaby, Canada,
2 WearTech Labs, Simon Fraser University, Burnaby, Canada

* smehdiza@sfu.ca, drsinamehdizadeh@gmail.com

## Abstract

We tested whether the same control principles that support steady-state walking are sufficient for robust, and rapid gait adaptations over a wide range of step lengths and frequencies. We begin by demonstrating that periodic gaits exist at combinations of step frequency and step length that span the full range of gaits achievable by humans. However, their open-loop stability is not enough to rapidly transition to target gaits. Next, we show that actuating with only one push-off and one hip spring of fixed stiffness cannot fully control the walker in the entire gait space. We solve this by adding a second hip spring with an independent stiffness to actuate the second half of the swing phase. This allowed us to design local feedback controllers that provided rapid convergence to target gaits by making once-per-step adjustments to control inputs. To adapt to a range of target gaits that vary over time, we interpolated between local controllers. This policy performs well, accurately tracking rapidly varying combinations of target step length and step frequency with human-like response times.

## Introduction

Effective walking requires adaptability. People rapidly and robustly adjust their gaits when searching for energy minimizing gaits, adjusting to an injury or new terrain, and learning how to take advantage of assistive devices [1–7]. Modeling and experimental studies of steady-state walking have shown that locomotion is primarily powered by two mechanisms: push-off by the stance foot at the end of the stance phase and active swing of the swing leg driven by hip muscles [8–12]. And a once-per-step control strategy—where push-off and hip actuation are adjusted only once per step—is sufficient to maintain steady-state walking and reject destabilizing perturbations [13–16]. Here we test whether these mechanisms that are used to power and stabilize steady-state gaits can be used to rapidly adapt to new gaits across the full range of step lengths and step frequencies observed in humans.

**Data availability statement:** There is no data associated with this publication. All code written in support of this publication is publicly available at: https://github.com/SFULocomotionLab/biped_models.

**Funding:** This work was funded by the Canadian Institutes of Health Research (SM, grant # 472824) and the Natural Sciences and Engineering Research Council of Canada (JMD, grant #ALLRP 586480-23). The funders had no role in study design, data collection and analysis, decision to publish, or preparation of the manuscript.

**Competing interests:** The authors have declared that no competing interests exist.

Once-per-step control is theoretically sufficient for rapid transitions across a wide range of gaits. Viability analysis by Zaytsev et al. [17,18] demonstrated that nearly any state within a model's viable region—the set of all states from which the system can avoid falling—can be guided toward a desired gait using only once-per-step control. Patil et al. [19] applied viability analysis to a compass walker and showed that regulating a single push-off impulse once per step can maintain the system within a recoverable speed range. Kelly and Ruina [20] and Patil et al. [21] both developed once-per-step controllers capable of adapting to different walking speeds. In the most comprehensive theoretical study of adaptation to date, Seethapathi et al. [22] used a simple model with once-per-step control to achieve adaptation across an impressively wide variety of walking tasks. Collectively, this body of theoretical work suggests that once-per-step control is not only sufficient for stabilizing steady-state gaits, but also for enabling adaptation between gaits. Other theoretical work has examined gait adaptation using more detailed neuromechanical models with continuous, reflex- or muscle-driven control that updates throughout the stride rather than once-per-step [23–25].

While these studies have substantially advanced our understanding of gait adaptation, opportunities remain to build on their foundational contributions. Here we extend prior work along three directions. First, we seek to understand the independent control of step length and step frequency—rather than speed alone—across the full range of human gaits. This distinction is important because the ability to adjust step length and step frequency independently enables more flexible gait adaptation. This is perhaps most clearly demonstrated when humans must adapt in speed-constrained gaits—such as adapting to a new assistive device during treadmill walking [1]. Second, we give the swing leg its own inertial dynamics and hip actuation, enabling active foot placement rather than prescribing footfall positions [22] or relying on passive motion [19,21]. This added realism reveals control challenges not present in simpler models, such as coordinating leg placement and timing under inertial coupling, and the exploration of gait adaptations that depend on independent swing-leg control. By actuating a dynamic swing leg, we open a richer and more challenging control space to study how once-per-step push-off and hip torque adjustments can span the full range of human gaits. Finally, we develop a framework for evaluating whether walking adaptation is occurring on human-like timescales and spanning the full space of step length and step frequency combinations achievable by humans.

Specifically, we tested whether the same principles of actuation and control shown to support steady-state walking are sufficient for robust, rapid gait adaptation over a wide range of step lengths and frequencies. We examine whether low-bandwidth actuation and control of both stance and swing legs can enable transitions between diverse gaits using minimal actuation and feedback, and accomplished within human-like time scales. To do so, we first demonstrate that periodic gaits exist across combinations of step frequency and length that cover the range of human gaits. We demonstrate that open-loop local stability is not enough to rapidly transition between gaits because some gaits in the gait space are unstable and that open-loop stable gaits have slow convergence rate. Next, we show having only two actuations—one

push-off and one hip spring of fixed stiffness cannot fully control the walker in all gaits across the entire gait space. We solve this by adding a second hip spring with an independent stiffness to actuate the swing leg in second half of the swing. We then design local feedback controllers to stabilize periodic gaits by making once-per-step adjustments to push-off and hip spring stiffnesses. We conclude by generalizing the local controllers into a control policy by interpolating control gains across a pre-determined grid of template gaits, allowing the controller to adapt to continuously changing target gaits.

## Materials and methods

### The walking model

We started with the powered simplest walker developed by Kuo [26] based on the earlier work of Garcia et al. [27] and McGeer [28]. It is a planar walking model with point feet and rigid legs connected by a frictionless hinge joint to a point-mass hip (Fig 1). There are no knee joints, nor is there an upper body. The legs are massless and the feet masses ($m$) are small relative to the hip mass ($M$) such that that the ratio $m/M$ approaches zero. This results in swing leg dynamics not affecting stance leg dynamics (but not vice versa). We studied two separate actuation methods for this model. In the first method, the original method suggested by Kuo [26], the actuation is provided by an impulse $P$ directed along the stance leg and a hip torque generated by a single hip spring with stiffness $k$, connecting the two legs. The push-off impulse $P$ is applied instantaneously just before the foot-ground contact. This contact is modelled as a perfectly inelastic collision that sets the initial conditions for the next step. In the second actuation method, the push off $P$ stays the same as the first method whereas the hip torque is generated by two independent hip springs with stiffnesses $k_1$ and $k_2$, each only active in respectively the first ($+\phi$) and second ($-\phi$) half of the swing (Fig 1). This means that the hip spring stiffness changes when swing angle crosses zero. We allow both the push-off impulse and hip stiffnesses to change between steps, but not within a step, resulting in the ability to control the walker once-per-step. Equations (1a-c) present the equations of motion for the stance and swing legs, along with the transition equations.

$$\ddot{\theta}(t) - \sin(\theta(t)) = 0 \tag{1a}$$

$$\ddot{\phi}(t) - \ddot{\theta}(t) - \dot{\theta}^2(t)\sin(\phi(t)) + \cos(\theta(t))\sin(\phi(t)) = \begin{cases} -k_1\phi(t), & \phi(t) > 0 \\ -k_2\phi(t), & \phi(t) \leq 0 \end{cases} \tag{1b}$$

$$\begin{bmatrix} \theta \\ \dot{\theta} \\ \phi \\ \dot{\phi} \end{bmatrix}^{+} = \begin{bmatrix} -1 & 0 & 0 & 0 \\ 0 & \cos(2\theta) & 0 & 0 \\ -2 & 0 & 0 & 0 \\ 0 & \cos(2\theta)(1-\cos(2\theta)) & 0 & 0 \end{bmatrix} \begin{bmatrix} \theta \\ \dot{\theta} \\ \phi \\ \dot{\phi} \end{bmatrix}^{-} + \begin{bmatrix} 0 \\ \sin(2\theta) \\ 0 \\ \sin(2\theta)(1-\cos(2\theta)) \end{bmatrix} P \tag{1c}$$

where $\theta$ is the stance angle defined with respect to vertical and $\phi$ is the swing angle defined with respect to the stance leg (Fig 1). All derivatives are with respect to time. The superscripts "-" and "+" denote states just before and after collision, respectively. We rendered all the parameters, equations, and results dimensionless by dividing masses by overall mass $M$, lengths by leg length $l$, and time by $\sqrt{l/g}$ where $g = 9.81\ ms^{-2}$ is the gravitational acceleration. Therefore, velocities are rendered dimensionless by dividing by $\sqrt{gl}$ and frequencies by dividing by $\sqrt{g/l}$. Note that for the single-spring walker $k_1 = k_2 = k$.

This walker has two independent degrees of freedom, represented by the stance angle ($\theta$) and stance angular velocity ($\dot{\theta}$) at the beginning of the step. The swing angle ($\phi$) and angular velocity ($\dot{\phi}$) at the start of the step are defined by the stance states and the push-off impulse using equation 1c. The reason for this dependence is two-fold: a) at foot ground contact, the swing angle is always twice the stance angle, and b) the swing leg mass in concentrated at the foot and thus

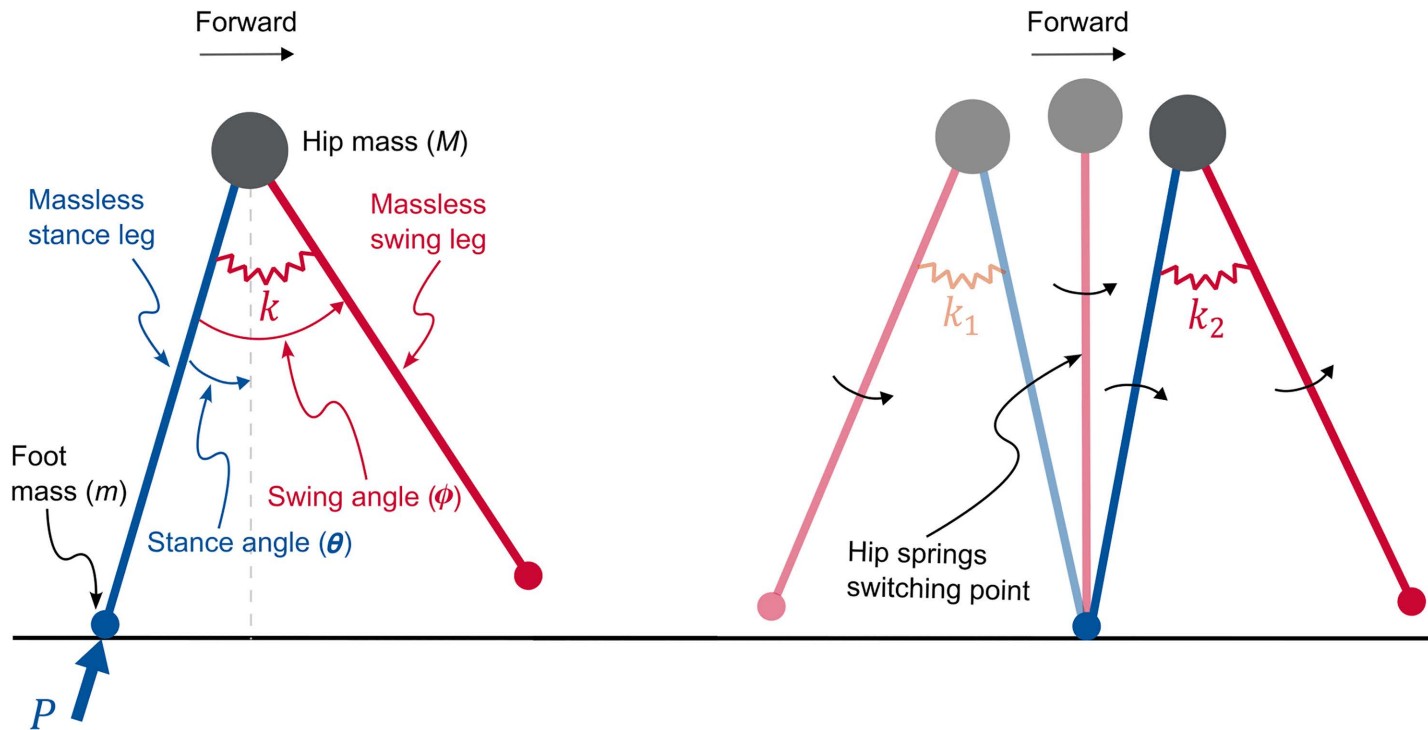

**A. Walker with a single hip spring**

Forward

Hip mass (*M*)

Massless stance leg

Massless swing leg

*k*

Foot mass (*m*)

Swing angle (*Φ*)

Stance angle (*θ*)

*P*

**B. Walker with two hip springs**

Forward

$k_1$

$k_2$

Hip springs switching point

**C. Angle and actuation time series**

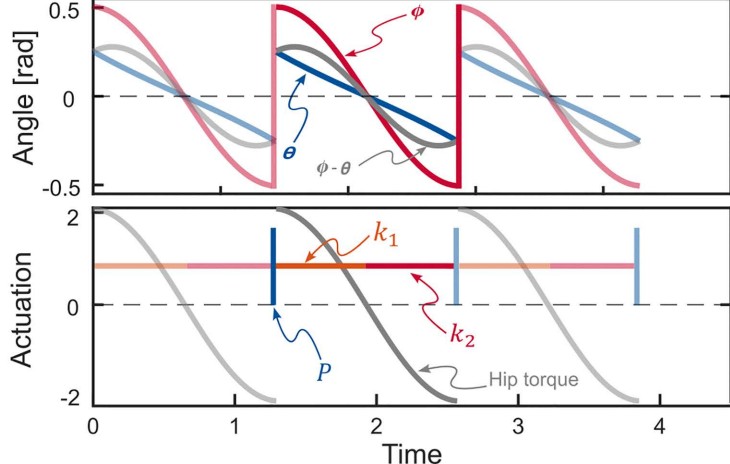

**Fig 1. Schematic representations of the walking models.** (A) The powered simplest walker with a single hip spring. It is a planar walking model with point feet and rigid legs connected by a hinge joint at the pelvis. There are no knee joints and upper body. Legs are massless and the feet masses are small relative to the center of mass resulting in swing leg dynamics not affecting stance leg dynamics. Actuation is provided by an impulse *P* directed along the stance leg and a hip torque generated by a hip spring with stiffness *k*, connecting the two legs. Stance leg angle (*θ*) is with respect to vertical and swing leg angle (*Φ*) is with respect to stance leg. (B) The same walker with two hip springs. Hip springs switch when swing leg angle (*Φ*) crosses zero. For a limit-cycle walking, that is when the stance leg is vertical, i.e., mid-stance (C) Angle and actuation time series for three walking steps of an exemplary gait. Angles are in radians and actuations are dimensionless.

makes no contribution to the angular momentum of the system about the contact point [27]. A step begins by specifying the four states—two independent (stance) and two dependent (swing) states—and applying hip torque to the swing leg and numerically integrating equations (1a) and (1b) until the end of the step. The step concludes when the swing foot contacts the ground (i.e., its vertical position reaches zero), at which point the instantaneous push-off and the transition in equation (1c) are applied to set the initial conditions for the next step.

## Periodic limit-cycle gaits

In this walking model, the state, $\bar{x} = [\theta, \ \dot{\theta}]^T$, at the beginning of a subsequent step, $\bar{x}_{i+1}$, can be considered a nonlinear function, $F$, of the state at the beginning of the previous step, $\bar{x}_i$:

$$\bar{x}_{i+1} = F(\bar{x}_i) \tag{2a}$$

A periodic limit cycle gait is a gait where the steps repeat identically:

$$\bar{x}^* = F(\bar{x}^*) \tag{2b}$$

Where $\bar{x}^*$ is the limit-cycle value of the state vector. We found limit-cycle gaits for step lengths and step frequencies within a dimensionless range of 0.1–1.1. For a leg length of one meter, this corresponds to step lengths of 0.1–1.1 meters and step frequencies of 0.31–3.44 steps per second (Hz), which encompass the full range of sustainable human walking capabilities, as previously studied by [29] (Fig 2A). Collectively, we refer to this set of gaits as the "gait space". For each combination of step length and frequency, we numerically solved equation 2b subject to both equality and inequality constraints to find the periodic gaits. The equality constraints included step length, step frequency, and limit-cycle conditions. The single inequality constraint was that push-off must be non-negative. We solved for the initial stance leg angle, initial stance angular velocity, push-off, and hip spring stiffness (or stiffnesses) that satisfied these constraints. We performed this analysis for all gaits in the gait space, totaling 10,201 limit-cycle gaits. The integration and solution tolerances were 1e-13 and 1e-11, respectively. We found the limit-cycle solutions for both the single and double hip spring cases, and verified that the double hip spring case found hip springs that were equal to each other and with the same magnitude as the single hip spring case. We refer to these periodic limit cycle gaits as "gaits" throughout this paper.

Finding limit-cycle solutions was computationally fast. We used trial and error to identify feasible initial guesses to find a limit cycle solution for the first gait we solved. This gait had the longest step length and fastest step frequency. We then computed all subsequent gaits systematically using warm-start continuation from neighboring solutions. To ensure consistency, we visually inspected the resulting parameter sets to verify that they varied smoothly across the gait space, with no discontinuities indicating that all of our solutions belonged to the same family of gaits. It took 251 seconds to find limit-cycle solutions for 94% of the 10,201 gaits on a desktop computer with a 32-core Intel processor (Dell, Core i9-13900 @3 Hz) equipped with a 44-gigabyte graphical processing unit (Nvidia GeForce RTX 3080 Ti). The automatic warm start method could not find the limit-cycle solutions for the remaining 6% gaits requiring manually trying multiple nearby solutions as the initial guess to find the solutions.

While we used numerical approaches to find the limit-cycle gaits, approximate analytical treatments of locomotion models—such as those developed for the Simple Linear Inverted Pendulum model [30–32] or Kuo's Idealized Simple Model [8]— could provide valuable intuition and can guide numerical analyses. In principle, our model could also be linearized in the regime of small stance angles, enabling approximate return maps that could generate initial guesses for numerical root-finding. However, because our study spans a wide gait space that includes large stance angles and strongly coupled swing–stance dynamics, such approximations would not remain valid across all conditions. For this reason, we relied on numerical methods to ensure comprehensive coverage of the full gait space.

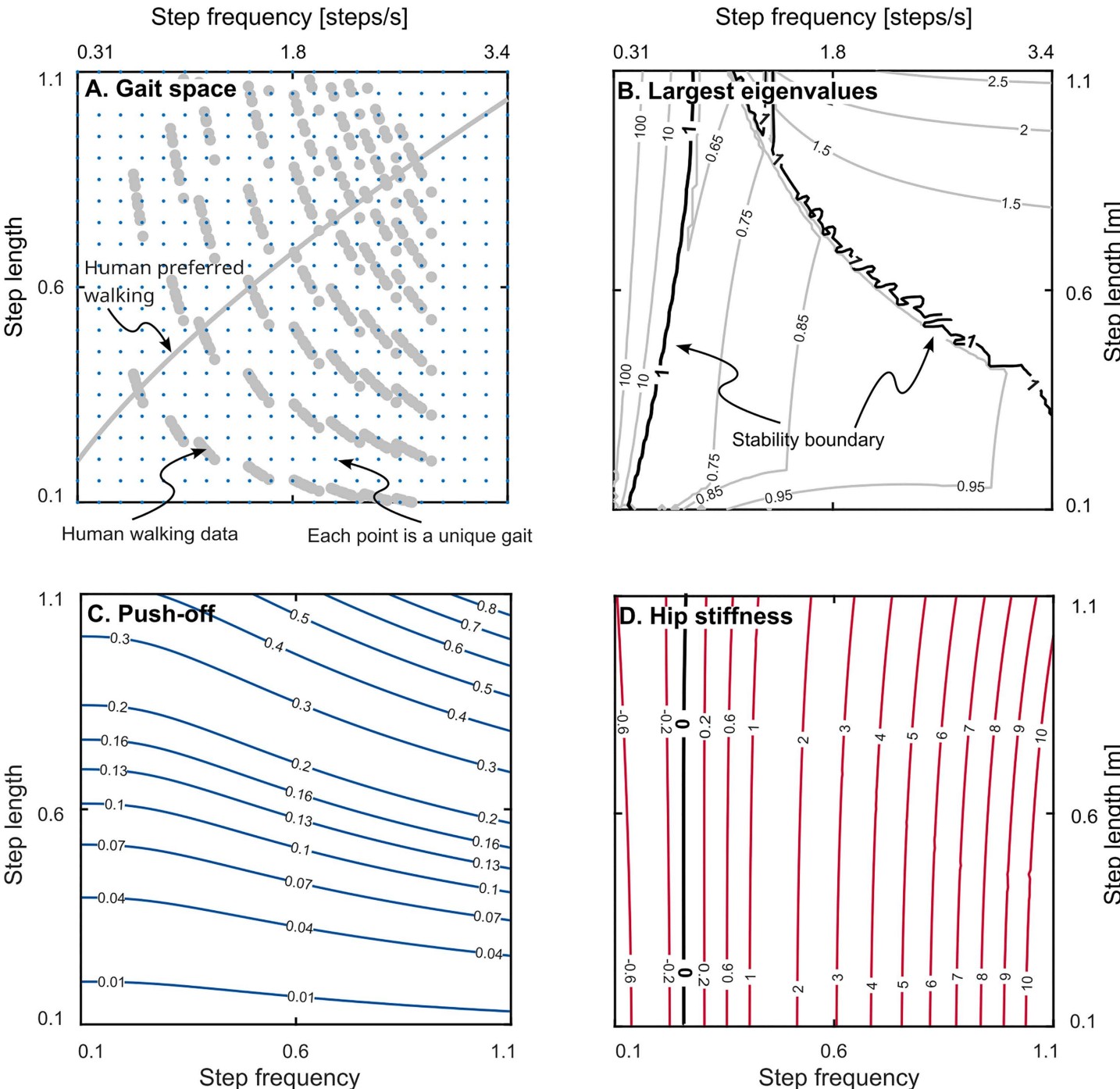

**Fig 2. Periodic limit-cycle gaits.** (A) the gait space. Each point on the graph is a limit-cycle gait determined by numerically solving the nonlinear equations of motion. The dark grey overlay shows the data from human experiment taken from Bertram [29]. The crossing line is human preferred walking. The graph shows that our gait space covers a wide range that goes beyond human walking capabilities. (B) Contours of the magnitude of the largest eigenvalues of matrix *A* (see Equation 3 in the text) for all gaits. The values below one show open-loop local stability of the uncontrolled walker. There is a stable region in the middle and two unstable regions at low step frequencies and combination of high step frequencies and high step length. These three regions are separated by two "stability boundaries". (C) contours of push-off impulse plotted for all gaits. It shows push-off changes minimally with step frequency especially in low step lengths. (D) contours of hip spring stiffness plotted for all gaits. It shows hip spring stiffness changes minimally with step length especially in low step frequencies. This plot is the same for both springs in the two-spring case. All the analyses were performed with dimensionless parameters. Left and bottom axes show dimensionless values and right and top axes show actual values.

## Stability analysis

We determined local open-loop dynamic stability (i.e., no active feedback control) for all gaits in the gait space. For each gait, we perturbed the two independent state variables ($\theta$ and $\dot\theta$) one at a time and determined how perturbations evolved over the next step. We set the perturbation size to 1e-7, which is sufficiently larger than the numerical tolerance of the equation of motion integration (1e-13) to avoid numerical artifacts, yet small enough to preserve the linearity of the return map. Mathematically, this is equivalent to building the discrete linear return map by linearizing around the limit-cycle:

$$\Delta\bar{x}_{i+1} = A\Delta\bar{x}_i \tag{3}$$

where $\Delta\bar{x}_{2\times1} = [\Delta\theta,\ \Delta\dot\theta]^T$, $A_{2\times2}$ is the Jacobian matrix that maps the deviations at the beginning of the current state to the next, $i$ is the step number, and $\Delta$ denotes deviation from the limit cycle values [28,33,34]. We determined the local stability by calculating the two eigenvalues of matrix $A$. If both eigenvalues have a magnitude less than one, the perturbations decay over subsequent steps, indicating local stability, otherwise the walker is locally unstable. The smaller the eigenvalues' magnitude, the faster the perturbations decay. Since the analysis is independent of the actuation method, the results hold for both actuation modes introduced above.

In addition to quantifying open-loop local stability, we determined the maximum perturbation size that the walker could tolerate from any direction without falling [27]. We performed this analysis for the open-loop walking case and when under active control (see below for the feedback controller design). For each target gait, we initialized the walker with the neighboring gait's initial conditions. In the open-loop walking case, the actuation was the target gait's limit-cycle values. In the active control case, the feedback controller determined the actuations. We repeated this process for all gaits in the nearest layer of neighboring gaits. If the walker successfully converged to the target gait and walked for 100 steps without falling, we expanded the search to the next layer of neighboring gaits. We continued this process until the walker failed to converge. For each target gait in the gait space, we defined the maximum tolerable perturbation as the maximum radius of the circle within which all neighboring gaits successfully converged to the target gait. We performed this analysis for all the gaits in the gait space (Fig 3).

## Controllability analysis

We performed controllability analysis of the walker in both single- and double-spring cases in the entire gait space by building the linearized discrete system dynamics for each gait:

$$\Delta\bar{x}_{i+1} = A\Delta\bar{x}_i + B\Delta\bar{u}_i \tag{4a}$$

$$\Delta\bar{y}_{i+1} = C\Delta\bar{x}_i + D\Delta\bar{u}_i \tag{4b}$$

where $\Delta\bar{x}_{2\times1} = [\Delta\theta,\ \Delta\dot\theta]^T$ similar to Equation (3), $\Delta\bar{u}_{2\times1} = [\Delta P,\ \Delta k]^T$ and $B_{2\times2}$, and $D_{2\times2}$ are the control input and direct transmission matrices, respectively, for the single spring case, $\Delta\bar{u}_{3\times1} = [\Delta P,\ \Delta k_1, \Delta k_2]^T$ and $B_{3\times2}$, and $D_{3\times2}$ are the control input and direct transmission matrices for the double-spring case, $\Delta\bar{y}_{2\times1} = [\Delta SL, \Delta SF]^T$, $C_{2\times2}$ is the sensor matrix for both single- and double-spring cases, and $SL$ and $SF$ are step length and step frequency, respectively. We conducted a formal controllability analysis by constructing the controllability matrix $\mathcal{C} = [B,\ AB]$ for each gait. The walker is controllable only if $\mathcal{C}$ is full rank [28,33,34].

## Local feedback controller

We developed a feedback controller for all gaits in the gait space. Following the controllability analysis, we focused exclusively on the double-spring walker and conducted all the subsequent analyses using this model (see the Results section).

**A. Basins of attraction without control**

**B. Basins of attraction with control**

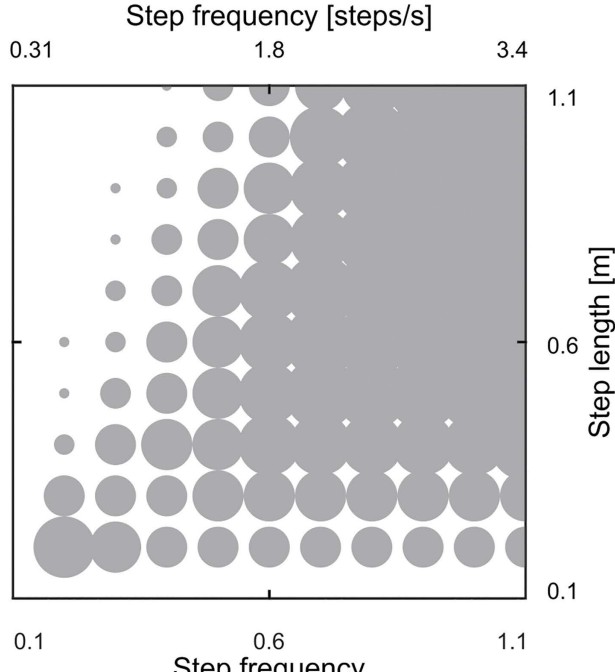

**C. Block diagram of local feedback controllers**

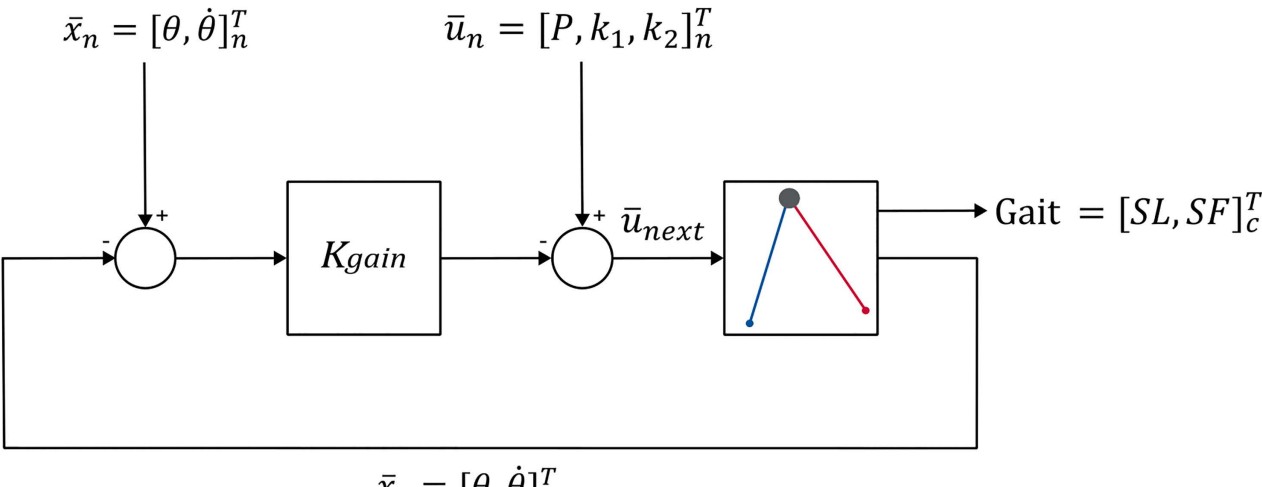

**Fig 3. Maximum tolerable perturbation analysis.** Maximum tolerable perturbation with (A) and without (B) feedback control for the double-spring walker plotted for every 10 gaits for better clarity. The bigger the size of the circles, the larger perturbation the walker can tolerate. Without a feedback controller, large areas of the gait space have near zero perturbation tolerance. The feedback controller increases the size of maximum tolerable perturbation for each gait and the overlap between the adjacent gaits which makes it easier to transition from one gait to another. (C) block diagram of the local feedback controller. The feedback controller computes the error between the current ($\bar{x}_c$) and nominal ($\bar{x}_n$) states of the walker and adjusts the nominal actuations ($\bar{u}_n$) for the next step ($\bar{u}_{next}$) in proportion to this error with proportional feedback gain $K_{gain}$. We determined feedback gains using pole placement method with poles at zero to eliminate local perturbations in one step. Left and bottom axes show dimensionless values and right and top axes show actual values. SL = step length, and SF = step frequency.

The state feedback controller used a proportional control law given by $\Delta \bar{u}_i = K_{gain} \Delta \bar{x}_i$, where $K_{gain}$ is the 3 × 2 gain matrix. Incorporating this into the return map equation (3), we obtained Equation (5):

$$\Delta \bar{x}_{i+1} = (A + BK_{gain}) \Delta \bar{x}_i \tag{5}$$

which is the feedback-controlled version of Equation (3) representing the return map for the controlled system. We used pole placement, with poles located at zero, to determine the controller gains, $K_{gain}$. This ensures that the eigenvalues of the matrix in parentheses are zero ensuring perturbations eliminate in a single step—also known as deadbeat control. We used pole placement to compute the feedback gain matrix for all gaits in the gait space resulting in a total of 10,201 unique gain matrices. The resulting local controller for each gait is defined as:

$$\bar{u}_{i+1} = \bar{u}_n - K_{gain}(\bar{x}_i - \bar{x}_n) \tag{6}$$

where the subscript "$n$" denotes the "nominal" limit-cycle values. The controller adjusts its actuations once per step just before foot contact where the push-off is applied.

## Performance of the local feedback controllers

While the maximum tolerable perturbation indicates the largest perturbation the walker can tolerate, it does not capture how quickly the walker eliminates perturbations and returns to the target gait. Furthermore, although our deadbeat feedback controller is designed to eliminate small disturbances in a single step, its effectiveness in diminishing larger disturbances cannot be inferred from a local linearized response. Therefore, in addition to evaluating the maximum tolerable perturbation under active control described earlier, we assessed the performance of the feedback controllers by computing their convergence response time—defined as the number of steps required for the walker to return to the target gait from a perturbed gait. We perturbed gaits away from the target gait in either step length or step frequency by an amount equal to the target gaits' magnitude of the maximum tolerable perturbation. For each target gait, we calculated four convergence response times, one for each of the positive and negative perturbations to step length and frequency. In each analysis, the walker began at the perturbed gait using the target gait's local feedback controller to compute the required actuations. We repeated this analysis for five representative target gaits within the gait space; gaits with dimensionless step length and step frequency pairs of [0.6, 0.6], [0.85, 0.85], [0.85, 0.35], [0.35, 0.85], and [0.35, 0.35] (Fig 4A). We found response time (also known as settling time) as the number of steps needed for the error between the current and target value to remain below 5% of the initial difference.

## General feedback policy

Using the above methods, we have 10,201 independent local feedback controllers for each of our grid gaits in the discretized gait space. But we seek a more general feedback policy that works for all gaits in our gait space, including gaits that are between grid points. To accomplish this, we used the grid of local controllers to interpolate control gains for between-grid gaits. This policy implements a 2-dimensional piece-wise linear interpolation function $G$, where the inputs are step length and step frequency, and the outputs are the six feedback gains:

$$K_{gain} = G(SL_t, \ SF_t; \ w_G) \tag{7}$$

Where $G$ is the interpolation function, $w_G$ is the parameters of the interpolation function, and subscript "$t$" denotes the target gait values. These target gaits can lie anywhere in the gait space and not just at one of the 10,201 grid points.

## A. Five representative target gaits

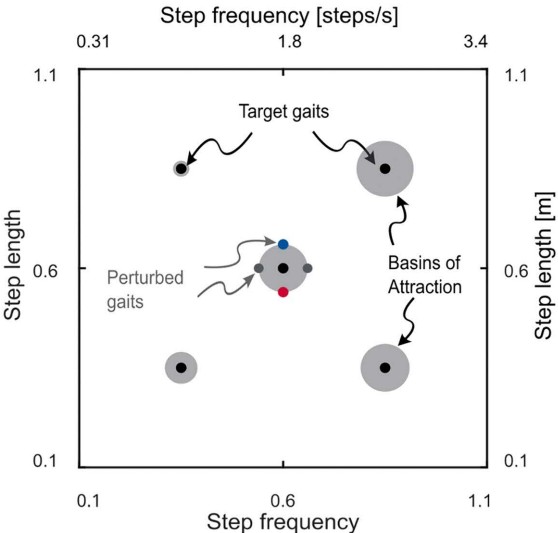

## B. Convergence graphs with local feedback controller

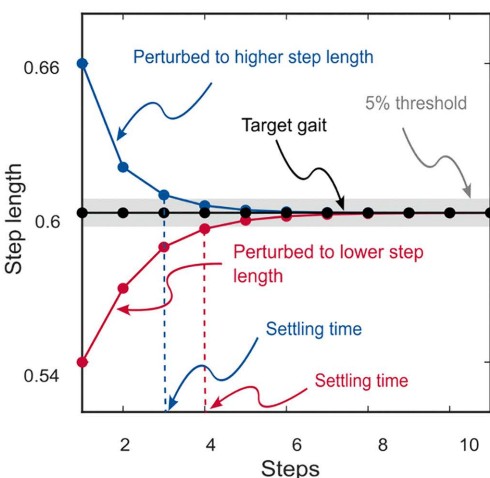

## C. Finding optimal convergence rate with control policy

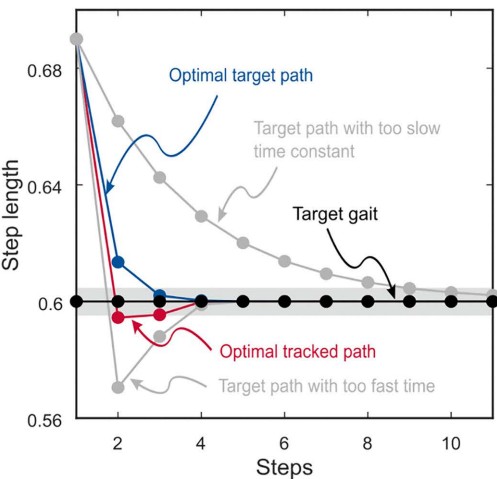

**Fig 4. Convergence rate analysis.** (A) Five representative target gaits and their maximum tolerable perturbation. The exemplary perturbed gait is also shown for one target gait. Perturbations were to step length and step frequency, and their magnitude was equal to the maximum tolerable perturbation of the target gait. (B) Convergence graph and settling times for two of the perturbed gaits in panel A. (C) Finding the optimal convergence rate with the feedback policy for one representative target point. The optimizer tries different time constants to find the optimal time constant. Left and bottom axes show dimensionless values and right and top axes show actual values.

## Performance of the feedback policy

We evaluated the performance of the feedback policy through two separate analyses. In the first, we computed the optimal response time of the feedback policy as it adapted from perturbed gaits to target gaits. This analysis allowed us to compare the feedback policy's performance to the response times observed in human walking. To enable this comparison, we aligned our analysis with the experimental design used in empirical studies of human gait adaptation [4,5]. That is, for each target gait, we perturbed step lengths 15% above and below the target gait's step length to specify two perturbed gaits. We then performed an optimization to determine the optimal response time required to converge (i.e., adapt) from the two perturbed gaits to the target gait. For this optimization, we designed two exponential reference paths that shared a time constant to be optimized. These paths began at the perturbed gaits whose step length deviated by ±15% from the target gait and approached the target gait over a pre-specified number of steps (see below). During each iteration of the optimization, the walker tracked these reference paths to converge to the target gait using the feedback policy. The cost function was thus the root mean square error between the actual tracked path and the reference path. The optimization's decision variable was the exponential time constant of the reference path. In other words, the optimization tried to find an optimal reference path whose tracking with the feedback policy had the lowest root mean square error among all possible reference paths. We repeated this process for perturbations to step frequency as well. As a result, each target gait had two optimal response times, one for step length and one for step frequency, that characterize its convergence speed. We performed this analysis for the five representative target gaits in the gait space described earlier to account for the differences in stability requirements of each region. We allowed a maximum of 10 steps for the reference path length in all target gaits except the target gait in the top left corner which required 30 steps when step frequency was perturbed.

In the second analysis, we assessed the feedback policy's performance in adapting to rapidly changing target gaits to test the walker's dynamic performance. We set the controlled walker to track several pre-specified paths in the gait space where step length and step frequency varied sinusoidally. Irrespective of path step count, the step length changed twice as often as step frequency, and the amplitude change of step length and step frequency was the same. As with the convergence rate analysis, we deployed these sine paths in five different regions of the gait space, where each path started from the representative gait as the initial gait. This accounts for the differences in stability requirements of each region. We chose these five points to capture a range of step length–frequency combinations while maintaining computational feasibility. In each case, the walker began from the limit-cycle state of the initial gait, reflecting the realistic scenario in which adaptation begins from an already stable gait rather than from a transient state. To test controller performance, we determined the minimum number of steps (i.e., minimum path length) required to complete the sinusoidal paths without falling. To track a path, the walker starts from the initial gait and uses the feedback policy to determine the actuations and controller gains required for the next step to converge to the next gait on the path. This process is repeated sequentially for all steps along the path. We reported the minimum number of steps required to complete each path without falling as well as the root mean square error between tracked and target step length and frequency as a percentage of these values at the starting point.

## Results

### Periodic gaits exist for a wide range of step lengths and step frequencies

We found periodic limit-cycle gaits for step lengths and step frequencies within a dimensionless range of 0.1–1.1. For a leg length of one meter, this corresponds to step lengths of 0.1–1.1 meters and step frequencies of 0.31–3.44 steps/s.

These ranges encompass and extend beyond the achievable gaits in human walking (Fig 2A). In total, we found a grid of 10,201 gaits (101 step lengths by 101 step frequencies in 0.01 increments). All limit cycle values of push-offs were positive meeting our positive push-off constraint (Fig 2C). Limit cycle values of hip spring stiffnesses changed sign approximately at the step frequency of 0.26 dimensionless (0.81 steps/s; Fig 2D). We refer to limit cycle values of push-off and hip spring stiffness as the nominal values.

## Open-loop stability is insufficient for gait adaptation

We found that periodic limit cycle gaits are unstable at very low step frequencies, and at combinations of high step frequencies and long step lengths (Fig 2B). This created two "stability boundaries" in the gait space: one at low frequencies (low frequency stability boundary) and another at a combination of high step frequencies and long step lengths (high frequency stability boundary; Fig 2B). The mechanism of instability can be described as follows. In our simple model of walking, energy is dissipated through the collision at the end of each step and the amount of dissipated energy increases with length of the step. Thus, to be stable, the walker must take a longer step in response to a perturbation that adds energy, and the length increase must be an appropriate amount. At the unstable fast step frequencies and long step lengths, an energy-adding perturbation shortens the step length rather than lengthening it. And at low frequencies, the lengthening of step length that arises in response to energy-adding perturbations is too long resulting in more energy dissipated than was added. While there is a large central region of the gait space that has open-loop dynamic stability, some of the eigenvalues within the stable region are close to one indicating that relying on open-loop stability to converge to new target gaits would be slow, taking hundreds of steps or more in some cases. Moreover, the maximum tolerable perturbation tends to be small—the walker would have to be close to a target gait to converge to it without falling (Fig 3A). Together, these findings highlight that open-loop stability is insufficient for gait adaptation and active feedback control is necessary for adaptation on human-like timescales.

It is not possible to control all gaits with a single push-off and a single hip spring with fixed stiffness Fig 5 presents contours of the values of the elements in the system dynamics and control input matrices $A$ and $B$, respectively, from the return map (See Equation 4a-b in the Methods). The contour plots illustrate that, for the single-spring walker, some matrix elements of equation (4a) are simultaneously zero along a specific line in the gait space which goes from gaits with medium step frequencies and high step lengths to gaits with high step frequencies and medium step lengths (the red lines in Fig 5). The presence of this "zero-boundary" line at the same gaits in both system dynamics, $A$, and control input, $B$, matrices demonstrates a complete lack of control over stance angle at these gaits for the single-spring walker. Mathematically, this is illustrated by expanding the first equation of the return map, equation (4a):

$$[\Delta\theta, \ \Delta\dot\theta]^T_{i+1} = \begin{bmatrix} A_{11} & A_{12} \\ A_{21} & A_{22} \end{bmatrix} [\Delta\theta, \ \Delta\dot\theta]^T_i + \begin{bmatrix} B_{11} & B_{12} \\ B_{21} & B_{22} \end{bmatrix} [\Delta P, \ \Delta k]^T_i \tag{4c}$$

Since $B_{12}$ is zero at the zero-boundary, hip stiffness $k_i$ cannot adjust stance angle in the next step ($\theta_{i+1}$) for these gaits. Push-off cannot modulate stance angle either because $B_{11}$ is zero for all gaits (push-off at the end of a step has no effect on stance angle at the beginning of the next step). Therefore, neither of the two actuations have direct control over the stance angle for gaits at the zero-boundary. This would not be an insurmountable control issue if there was dynamic coupling between states where a change in stance angular velocity due to a change in push-off in the present step resulted in change in stance angle in the next step ($A_{21}$). This dynamic coupling does happen throughout much of the gait space but disappears for the gaits that lie on the zero-boundary (Fig 5). This complete loss of control for gaits at the zero-boundary is also reflected in the rank deficient controllability matrix $\mathcal{C} = [B, \ AB]$. Since $A_{21}$, $B_{21}$, and $B_{22}$ are zero for these gaits, one column of the 2x2 $\mathcal{C}$ will be zero showing loss of control. In summary, only a single push off and a single hip spring of fixed stiffness—typically used to control the walker in steady state walking—cannot fully control the simplest walker across all gaits in the gait space.

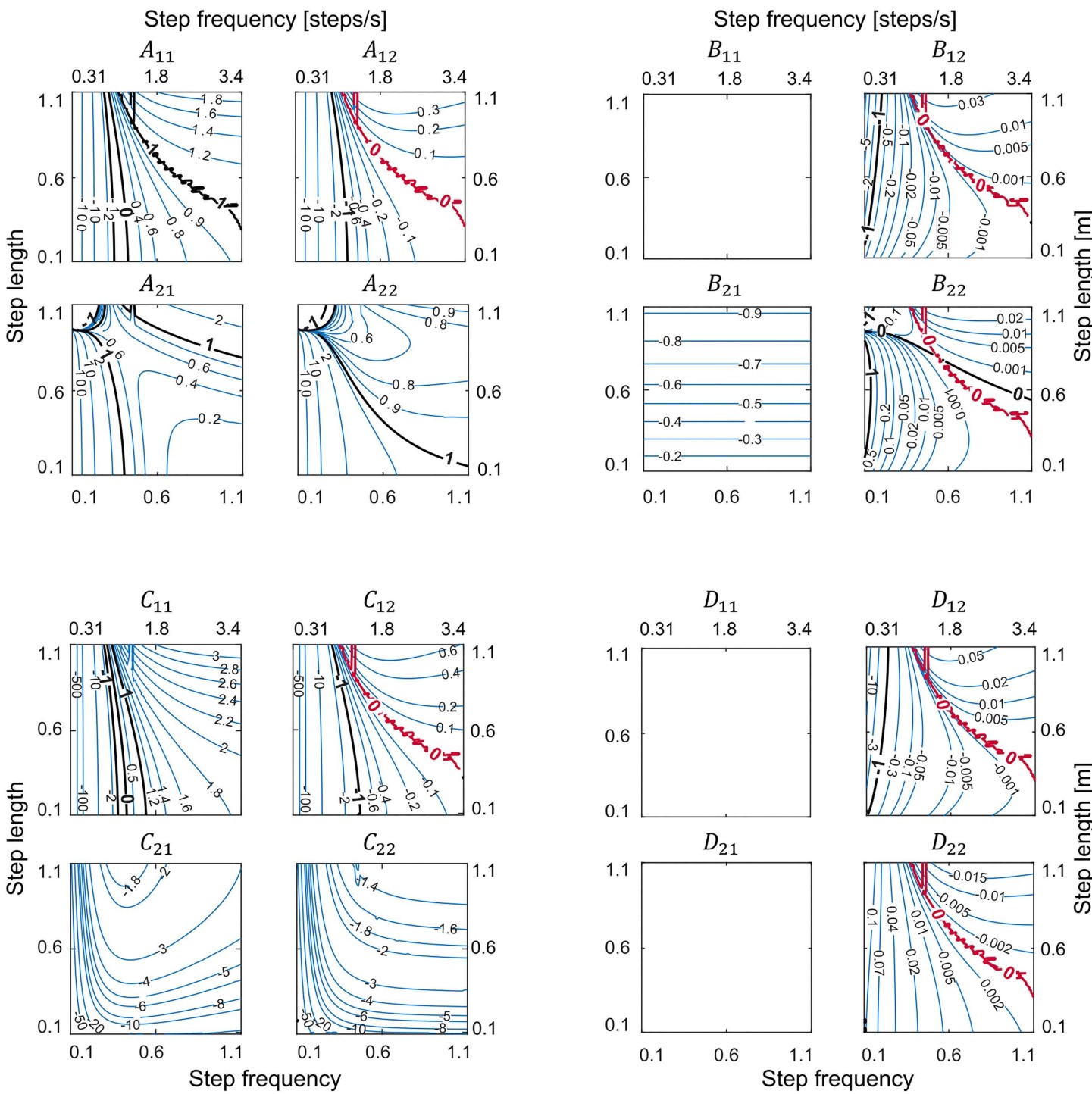

**Fig 5. Controllability analysis for the single-spring walker.** Contour plots of components of system dynamics *(A)*, control input *(B)*, output *(C)*, and feedthrough *(D)* matrices from the linear return map for all gaits in the gait space (see equations 4a-4b in the text). Matrices *A* and *C* are the same for the single and double-spring walkers. Matrices *B* and *D* are for the single-spring walker only. The existence of a common zero-boundary (red line) in $A_{12}$, $B_{12}$, and $B_{22}$ results in gaits along them being uncontrollable. These zero-boundaries coincide with the right stability boundary in Fig 2B. Components $B_{11}$, $D_{11}$, and $D_{21}$ are zero for all gaits because push-off cannot affect stance angle, step length or step frequency. Left and bottom axes show dimensionless values and right and top axes show actual values.

## Adding a spring to achieve control over the entire gait space

The results of the previous section demonstrated that a single hip spring loses control of the swing leg in adjusting the stance angle for gaits on the zero-boundary. We hypothesized that introducing a second independent hip actuation could compensate for this loss, allowing for precise adjustment of the stance angle for the next step. While this second hip actuation could take various forms—such as motors, muscles, or impulsive torques—a simple approach is to add a second spring with a separate stiffness from the first one. In this actuation method, only one spring is active at a time—the initial spring in the first half of the swing phase (positive swing angle) and the final spring in the second half (negative swing angle). This means that the hip spring stiffness changes when the swing angle crosses zero. The two hip springs have identical stiffnesses when periodic walking, and their values are identical to the limit cycles for the single spring model. Consequently, the local stability analysis presented earlier remains unchanged with the addition of the second identical spring because matrix $A$ determines how disturbances to state variables evolve over time when actuations are kept constant.

As a result of adding a second hip spring, the controllability matrix $\mathcal{C}$ becomes full rank indicating full control. Inspecting the contour plots of the control input matrix $B$ (equation 4a; Fig 6) indicates that, although zero-boundaries still exist in some of the components that determine the effect of the two spring stiffnesses on walker state, these zero-boundaries are at different gaits between the elements of $B$. This ensures that one of the two springs always retains control authority over stance angle irrespective of dynamic coupling. While these two springs can also influence stance angular velocity, push-off always has control authority over this state. In summary, we achieve full control over the entire gait space with three

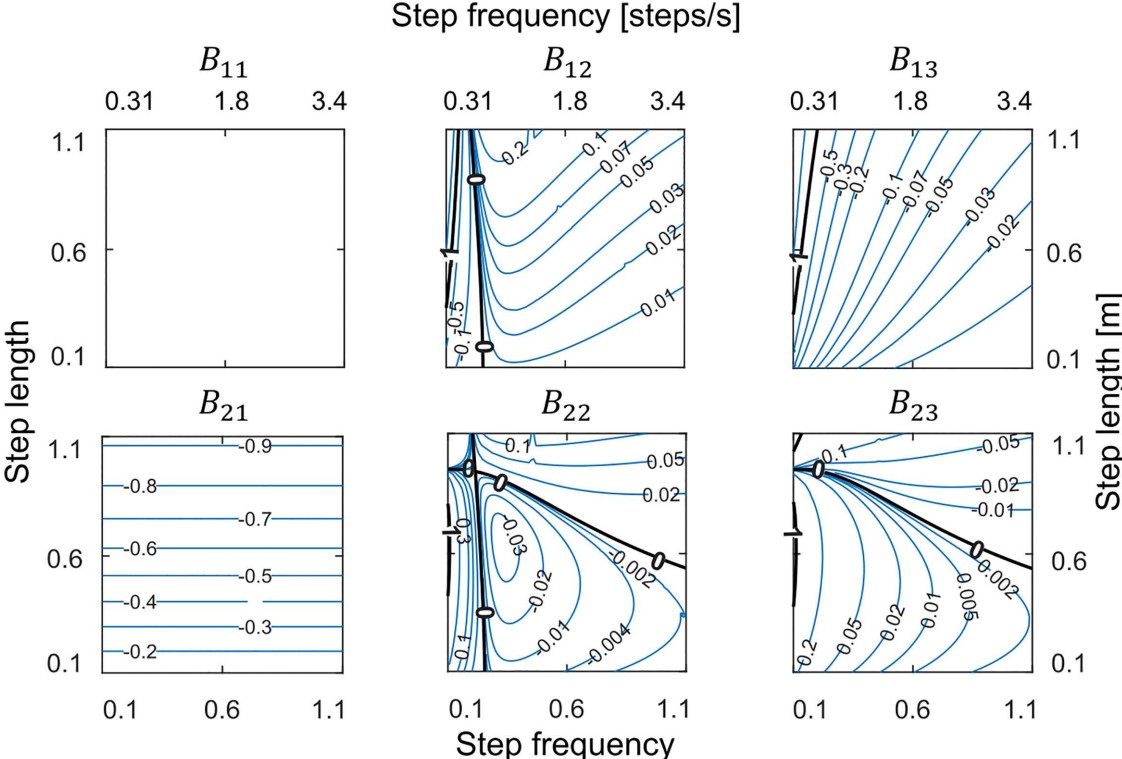

**Fig 6. Controllability analysis for the double-spring walker.** Contour plots of components of the control input matrix $B$ for the two-spring walker. There is no common zero-boundary with $A_{12}$ (see Fig 5) which shows regaining control for all gaits by adding the second hip spring. Left and bottom axes show dimensionless values and right and top axes show actual values.

actuations—a push-off, an initial hip spring, and a final hip spring—that change once per step. Thus, we proceed with this new actuation mode for the remainder of the analyses.

## Once-per-step local feedback control provides high performing control over periodic gaits

We first designed separate local feedback controllers for each limit-cycle gait on the grid. These deadbeat controllers (Equation 6 and Fig 3C) stabilized all previously open-loop unstable gaits in response to small disturbances and increased the local convergence rate across the entire gait space. Specifically, in the passively stable region in the center, the mean largest eigenvalues were reduced from ~0.8 without active control to zero with active control. In the open-loop unstable regions on the left and right, the mean largest eigenvalues were reduced from ~10,000 and ~1.6, respectively, to zero with active control. These local feedback controllers also increased the maximum tolerable perturbation for all gaits (Fig 3), enabling the walker to tolerate larger disturbances. The average size of the maximum tolerable perturbation increased eight-fold from $0.005 \pm 0.011$ (dimensionless) without the controller to $0.045 \pm 0.027$ with the feedback controller.

While the maximum tolerable perturbation indicates the largest perturbation the walker can tolerate, it does not capture how quickly the walker eliminates perturbations and returns to the target gait. We thus determined the response time of the local feedback controllers when converging to target gaits from perturbed gaits. After the human experiment work of Snaterse, Pagliara and colleagues [4,5], we define response time as the number of steps needed for the error between the current and target value to remain below 5% of the initial difference. Here we perturbed step length and step frequency with a perturbation size equal to the size of the maximum tolerable perturbation as it is the largest perturbation that local controllers could reliably reject. We performed this analysis for five representative gaits in the gait space (Table 1 and Fig 4). The response time to converge to target gaits was fast for all five representative target gaits when under local feedback control, averaging $2.5 \pm 0.6$ steps and $4.9 \pm 1.4$ steps when we perturbed step length and step frequency, respectively.

## A feedback policy for adaptation

We generalized the local feedback controllers into a more general feedback policy by interpolating control gains across the pre-determined grid of template gaits. This policy implements a 2-dimensional piece-wise linear interpolation function where the inputs are step length and step frequency, and the outputs are the six feedback gains. To compare the performance our control policy to human experiments, we evaluated the optimal convergence response time of this policy to target gaits whose step lengths or frequencies were 15% above or below the current gait (Fig 4C). This 15% perturbation is larger than the perturbation size we used for evaluating the local feedback controllers, and equal to that used in human studies [4,5]. We found that the optimal response time was within five steps for all representative gaits except for the gait in the high step length and low step frequency region of which perturbation to step frequency required a response time 11

Table 1. Response time of the local feedback controller for five representative gaits presented in units of walking steps. The perturbation magnitude was equal to magnitude of the maximum tolerable perturbation (MTP) of each gait. SL = step length, and SF = step frequency. Step length, step frequencies and MTP are dimensionless.

| Region | Gait [SL, SF] | MTP magnitude | Increased SF | Decreased SF | Increased SL | Decreased SL |
|---|---|---|---|---|---|---|
| | | | Response time (steps) | | | |
| **Centre** | [0.60, 0.60] | 0.06 | 5.2 | 4.1 | 2.8 | 2.6 |
| **Top right** | [0.85, 0.85] | 0.07 | 4.9 | 4.2 | 2.9 | 2.7 |
| **Top left** | [0.85, 0.35] | 0.02 | 8.2 | 6.1 | 1.9 | 1.9 |
| **Bottom right** | [0.35, 0.85] | 0.06 | 3.9 | 3.6 | 3.8 | 2.7 |
| **Bottom left** | [0.35, 0.35] | 0.04 | 4.9 | 3.9 | 1.9 | 1.9 |

steps to converge (Table 2). This well-matches human experiments, which reported response times ranging from two to six steps following perturbations to step frequency and speed [5].

We also evaluated the performance of this general feedback policy to adapt to paths of varying step lengths and step frequencies (Figs 7 and 8). The walker successfully tracked sine-shaped paths in all regions of the gait space, demonstrating its ability to adapt to varying step lengths and frequencies. However, tracking performance varied across regions (Table 3). The walker could complete the path in the high step length and high step frequency (top right) region with as low as 18 steps while it needed 191 step to complete the path in the long step length and low step frequency (top left) region. The poorer path-tracking performance in top left region is consistent with the longer response times we found in that region (Table 2).

## Discussion

We demonstrated that the same method of actuation and control of steady-state walking are insufficient for robust and rapid gait adaptation over a wide range of step lengths and frequencies. Specifically, we showed that relying on only one push-off and one hip spring of fixed stiffness cannot fully control the walker in the entire gait space. Thus, we added a third actuation, allowing the controller to adjust the hip spring stiffness midway through the swing phase. This allowed us to design feedback controllers that stabilized periodic gaits by making once-per-step adjustments to push-off and hip spring stiffnesses providing convergence to target gaits within human-like time scales. Finally, we extended the controller to handle rapidly changing target gaits by interpolating control gains across the grid of periodic gaits.

Our work has several limitations. One limitation is the generalizability to more complex walking models or robots—it may prove difficult to find limit-cycle solutions across a wide range of gaits in 3D bipeds or those with knee joints or an upper body. We suspect that this is not a fundamental issue as warm-start methods can extend existing solutions to a broader range of gaits [33,35,36]. Another potential limitation is that our linear feedback controller is not always able to handle very large and rapid changes to target gaits. We suspect this challenge can be addressed by refining the feedback gains with optimization techniques to enhance robustness to larger disturbances [13]. We also assumed full state feedback, no sensory noise, and no actuator limits (except for the positive push-off constraint). While our assumptions may not fully reflect human and robotic walking, they do not fundamentally limit our approach. Our results can be extended to account for these factors using established frameworks such as Kalman filters and linear quadratic regulators [13,14,37].

Why are the stability boundary gaits in the single spring model also the gaits at which control authority is lost? Modulating stance angle is essential for control in this walker as it is only through changes in step length that energy is dissipated. The state of the walker at the end of a step is determined by both the stance leg dynamics (how big of an angle it sweeps and the speed at which it sweeps) as well as the swing leg dynamics (how far it must swing forward and backward and the speed at which it accomplishes this). For stable gaits with medium step frequency range (gaits in the central region of the gait space), increasing stiffness speeds the swing leg such that after it swings forward further it then swings backward

**Table 2. Optimal response time of the control policy expressed in number of walking steps for perturbations to step length (SL) and step frequency (SF). The perturbation size was 15% below and above the target gait step length or step frequency. Step length and step frequencies are dimensionless.**

| Region | Gait [SL, SF] | Perturbation to step frequency | Perturbation to step length |
|---|---|---|---|
| | | Response time (steps) | |
| Centre | [0.60, 0.60] | 4.0 | 1.6 |
| Top right | [0.85, 0.85] | 4.7 | 2.2 |
| Top left | [0.85, 0.35] | 11.0 | 1.6 |
| Bottom right | [0.35, 0.85] | 3.5 | 1.8 |
| Bottom left | [0.35, 0.35] | 3.1 | 0.8 |

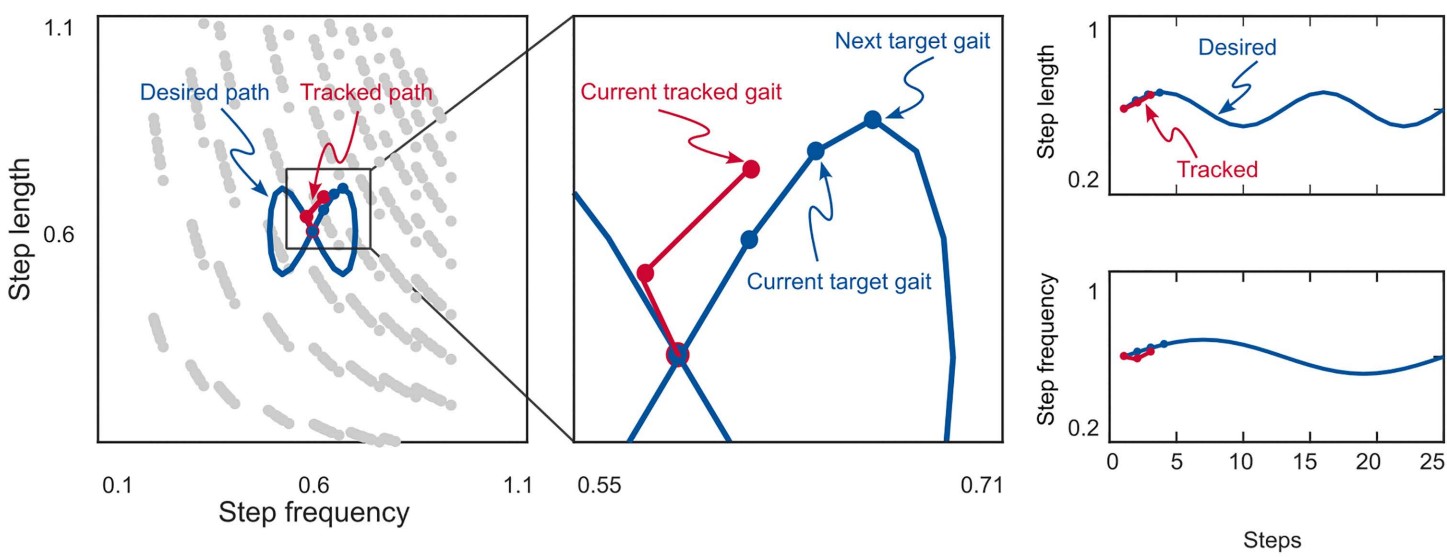

**A. Tracking of a representative path**

**B. Zoomed view**

**C. Time series**

**D. Control block diagram for tracking**

**E. Pseudocode for tracking algorithm**

Input: a discretized path to be tracked
Initialize the walker with the states of the first gait on the path as the *current gait*.
Loop for each gait on the path:
  Set the next consecutive gait on the path as the *target gait*.
  Use table lookup or interpolation to determine *target gait's* states ($\bar{x}_t = [\theta, \dot{\theta}]_t^T$), nominal actuations ($\bar{u}_t = [P, k_1, k_2]_t^T$), and feedback gains ($K_{target}$).
  Define the feedback controller's parameters as:
    $\bar{x}_n = \bar{x}_t$
    $\bar{u}_n = \bar{u}_t$
    $K_{gain} = K_{target}$
  Calculate next actuations as: $\bar{u}_{next} = \bar{u}_n - K_{gain}(\bar{x}_c - \bar{x}_n)$
  Apply $\bar{u}_{next}$ to the walker, integrate for one step and record the walker's *next state*.
  Calculate and record step length and step frequency.
  *current gait = next gait*

**Fig 7. Gait adaptation analysis.** (A) Representation of an example sine path with four individual gaits (blue circles) along the path being tracked by the walker (red circles). (B) Zoomed in insert showing current and target gaits. (C) Step length and step frequency time series of the path plotted separately. (D) Block diagram of the feedback policy. Feedback policy uses a piece-wise linear interpolation to determine feedback gains for each target gait along the path. (E) Pseudocode for the tracking algorithm. Left and bottom axes show dimensionless values and right and top axes show actual values. See the caption of Fig 3 for notations.

further, contacting the ground earlier. This decreases step time (Fig 5, $D_{22}$), step length ($D_{12}$), and stance angle ($B_{12}$). For unstable gaits to the right of the right stability boundary, increasing stiffness has the opposite effect—the forward swing of the foot extends the time it takes for the foot to contact the ground (Fig 5). This increases step length and increases stance angle at contact. And at intermediate gaits, there is a transition between these opposite effects such that changing swing leg dynamics by increasing stiffness has no effect on step time, step length or stance angle. It is precisely these interme-diate gaits where changing stance dynamics by slowing stance angular velocity (via a change to $\Delta\dot{\theta}$) also has no effect on

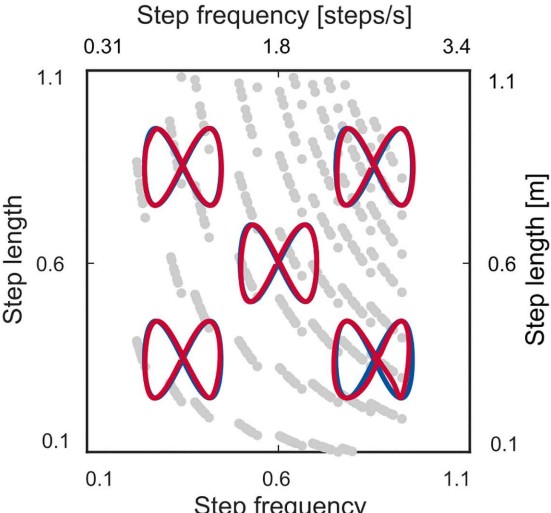

**Fig 8. Gait adaptation in five representative sine paths in different regions of the gait space.** (A) each path has the minimum number of steps. See Table 3 for the minimum number of required steps for each path. (B) each path has 100 steps per path except for the top left path which is with 200. Left and bottom axes show dimensionless values and right and top axes show actual values.

**Table 3. Performance of the feedback policy in tracking the sine paths in five representative regions of the gait space. SL = step length, SF = step frequency, RMS = root mean squared. Step length and step frequencies are dimensionless. RMS errors are presented as the percentage of the step length and frequency of the starting gait of the path.**

| Region | Starting gait [SL, SF] | Minimum number of steps | Mean RMS error ([%SL, %SF]) |
|---|---|---|---|
| **Centre** | [0.6, 0.6] | 22 | [14.95, 33.58] |
| **Top right** | [0.85, 0.85] | 18 | [14.17, 31.69] |
| **Top left** | [0.85, 0.35] | 191 | [0.96, 11.06] |
| **Bottom right** | [0.35, 0.85] | 33 | [19.42, 45.80] |
| **Bottom left** | [0.35, 0.35] | 33 | [24.34, 43.91] |

step length ($C_{12}$) or stance angle ($A_{12}$). More simply, slowing stance dynamics produces a similar effect to speeding swing dynamics, and at intermediate gaits, neither is effective at modulating step length—a necessary condition for controllability. Using separate hip spring stiffnesses for the two halves of the swing phase decoupled their dynamics making it possible to modulate the second stiffness to target any step length irrespective of what transpired in the first half of swing.

We also explored alternative swing leg actuation strategies beyond the two-spring approach. Specifically, we added a damper in parallel with the single hip spring [38] and found that this modification restored control authority over gaits that were previously unstable. Notably, this configuration has an appealing feature: the limit-cycle solution itself involves zero damping—relying solely on push-off and a single hip spring—while damping becomes active only in response to perturbations. This mirrors the behavior of our two-hip-spring model, where both springs have identical stiffness at the limit cycle and differ only when the walker is perturbed. More generally, a variety of actuation strategies could achieve stability, provided they allow for independent modulation of stance and swing dynamics under perturbation. We favor the two-hip-spring model because it introduces minimal structural complexity while offering a straightforward biomechanical interpretation—corresponding to independent hip flexor and extensor torques active during early and late swing, respectively.

We found gaits with long step lengths and low step frequencies challenging to control. In this region of the gait space, small perturbations that increase system energy led to disproportionately large increases in step length, dissipating far more energy than the original perturbation added ($C_{11} \ll 0$ and $C_{12} \ll 0$; Fig 5). The underlying physical intuition is that at low step frequencies, there is more time for perturbations to grow and influence step timing. And at long step lengths, small variations in foot-ground contact timing produce large changes in step length. Consequently, the controller gains required for stabilization will depend strongly on step length and step frequency leading to small maximum tolerable perturbation in this region (Fig 3). This sensitivity of controller gains also poses a challenge for our approach to generating a feedback policy, as small interpolation errors in controller gains can lead to significant instability. People report difficulty in executing walking gaits with these step length and step frequency combinations [29], perhaps for the same reasons that we had difficulty in controlling our walker.

Our results on stability after control (both maximum tolerable perturbation and the tracking performance) are based on using a linear deadbeat controller. Alternative controller design approaches—such as linear quadratic regulation (LQR), nonlinear controller design, or optimization-based methods—could produce different outcomes. We chose the deadbeat approach because it enables the walker to eliminate disturbances in a single step and avoids the computational overhead associated with some of these other methods. Further, our preliminary experiments did not reveal significant performance differences between the deadbeat controller and an LQR-based controller. Importantly, our design approach—offline computation of controller gains combined with online interpolation—is independent of the specific controller type and can be extended to incorporate other control strategies.

Our findings may provide insights into human locomotion control. One insight originates from our once-per-step control approach. The once-per-step control eliminates reliance on pre-planning gait trajectories. Instead, the controller adapts in real-time, dynamically adjusting actuations based on the immediate target gait. This still performed within human-like time scales of adaptation suggesting that humans could also take advantage of once-per-step control for effective adaptation of walking gaits [4,5]. We found that it is essential for control to have independent modulation of the hip torque between the first and second half of swing. This might explain the pattern of hip muscle activity during the swing phase of human walking where flexor muscles accelerate the swing leg in the first half, and extensors muscles decelerate it in the second half of the swing phase [39].

Our controller design is computationally simple and efficient. Two key design features contribute to the simplicity of our method. First, the control operates discretely, adjusting actuations once per step rather than continuously. Since step durations tend to be significantly longer than the computational bandwidth of both modern computers and sensorimotor control systems of humans and animals, this discrete approach is far less computationally demanding than continuous control within a step [14]. Second, most computations are performed offline, with online computations limited to linear interpolation. Among the offline computations, finding limit-cycle gaits is the most computationally intensive step. This arises from the challenge of selecting a feasible initial guess for optimization, which often requires trial and error. With a suitable initial guess for the first gait, and using warm start methods, finding solutions was fast much like how a human infant quickly learns new ways to walk after taking months to find their first steps. Once we found periodic gaits, our method for determining the local controller gains only required matrix algebra. This is computationally faster than previous approaches that relied on nonlinear optimization for real-time control and adaptation of biped robots [19,20,40–43]. Although the computational demands of our offline methods will increase with degrees of freedom, the efficiency of real-time interpolation remains a key advantage.

Our rapidly converging feedback policy can be part of a hierarchical control framework to model, and perhaps understand, human gait adaptation. In this framework, a high-level controller dynamically changes the target gait towards an optimal performance objective and then relies on a low-level policy like ours to rapidly and stably guide the walker to the target gait [22–25]. For example, an objective of a walking human is to minimize energy [1,22,44–46]. In this hierarchical control approach, at each walking step the higher-level controller senses energetic cost and assigns a new target gait to

the lower-level controller. The new target might allow the walker to exploit lower cost gaits or explore new gaits, but either way, the higher-level controller can rely on the lower-level policy to fulfill its instructions and rapidly adapt to a wide range of target step lengths and frequencies.

## Acknowledgments

We would like to thank David Remy for reading the initial draft of the manuscript and providing helpful comments. We also thank Sam Burden and Varun Joshi for their insightful comments on our results and their helpful discussions throughout the development of this work.

## Author contributions

**Conceptualization:** Sina Mehdizadeh, James Maxwell Donelan.

**Data curation:** Sina Mehdizadeh, James Maxwell Donelan.

**Formal analysis:** Sina Mehdizadeh, James Maxwell Donelan.

**Funding acquisition:** Sina Mehdizadeh, James Maxwell Donelan.

**Investigation:** Sina Mehdizadeh, James Maxwell Donelan.

**Methodology:** Sina Mehdizadeh, James Maxwell Donelan.

**Project administration:** James Maxwell Donelan.

**Resources:** Sina Mehdizadeh, James Maxwell Donelan.

**Supervision:** James Maxwell Donelan.

**Validation:** Sina Mehdizadeh, James Maxwell Donelan.

**Visualization:** Sina Mehdizadeh, James Maxwell Donelan.

**Writing – original draft:** Sina Mehdizadeh, James Maxwell Donelan.

**Writing – review & editing:** Sina Mehdizadeh, James Maxwell Donelan.

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
