## [Decision Letter · Decision Letter 0]

3 Sep 2025

Dear Dr. Mehdizadeh,

Thank you for submitting your manuscript to PLOS ONE. After careful consideration, we feel that it has merit but does not fully meet PLOS ONE’s publication criteria as it currently stands. Therefore, we invite you to submit a revised version of the manuscript that addresses the points raised during the review process.

While the reviewers were generally positive about the study, there are some points where additional discussion are needed. The NN section in particular needs to be strengthened and the reviewers give some suggestions about how to do so.

We look forward to receiving your revised manuscript.

Kind regards,

Anne E. Martin

Academic Editor

PLOS ONE

Journal Requirements:

2. Please update your submission to use the PLOS LaTeX template. The template and more information on our requirements for LaTeX submissions can be found at http://journals.plos.org/plosone/s/latex .

Reviewers' comments:

Reviewer's Responses to Questions

**Comments to the Author**

1. Is the manuscript technically sound, and do the data support the conclusions?

Reviewer #1: Yes

Reviewer #2: Yes

Reviewer #3: Partly

2. Has the statistical analysis been performed appropriately and rigorously?

Reviewer #1: Yes

Reviewer #2: N/A

Reviewer #3: Yes

3. Have the authors made all data underlying the findings in their manuscript fully available?

Reviewer #1: Yes

Reviewer #2: Yes

Reviewer #3: Yes

4. Is the manuscript presented in an intelligible fashion and written in standard English?

Reviewer #1: Yes

Reviewer #2: Yes

Reviewer #3: Yes

Reviewer #1: PONE-D-25-37501 Review

Thank you for the opportunity to review this interesting paper.

-----

Here are some comments based on PLOS ONE criteria.

1. Original Research

The manuscript presents new modeling and control frameworks for gait adaptation, analyzes the limits of standard actuation methods, proposes and tests a novel two-stage hip actuation policy, and evaluates a neural network-based controller. I believe this is original research and I am not aware of papers dealing with the same analysis.

2. Results Not Published Elsewhere

A version of this paper is available as a preprint on bioRxiv, but as far as I know, it is not published in a scientific journal.

3. Technical Quality and Detail

The study includes mathematical modeling, stability and controllability analyses, and simulations. Model details, optimization routines, mathematical formulations, and simulation conditions are presented in most parts (see below) clearly and in detail (e.g., equations, algorithms, parameter values, methods for generating figures and tables).

The code used for the analysis is made publicly available at:

GitHub - SFULocomotionLab/biped_models.

This is very much appreciated.

4. Conclusions Supported by Data

The primary conclusions [namely that standard control is insufficient for robust gait adaptation, and that adding a second hip spring with an independent stiffness to actuate the second half of the swing phase is a way to solve the problem] are backed by computational experiments.

5. Intelligible English

The manuscript is well-written in academic English. The abstract, author summary, introduction, discussion, methods, and references are clearly organized.

6. Ethical and Integrity Standards

This is a computational and modeling study, so human or animal ethics approval is not required, I believe. Competing interests and funding sources are fully disclosed.

-----

Here are some comments about the content of the manuscript.

Model.

I feel that some analytical work on your model 1a,1b,1c can be done, instead of relying only on numerics. At the end, the angle swept during stance should be relatively easy to compute especially under some parameter regimes and this should be enough to determine, at least, an approximate apex return map for the model. See for example what is done in these papers:

Geyer H, Seyfarth A, Blickhan R. Spring-mass running: simple approximate solution and application to gait stability. J Theor Biol. 2005 Feb 7;232(3):315-28. doi: 10.1016/j.jtbi.2004.08.015. PMID: 15572057.

Selvitella AM, Foster KL. An approximate solution of the SLIP model under the regime of linear angular dynamics during stance and the stability of symmetric periodic running gaits. J Theor Biol. 2024 Dec 7;595:111934. doi: 10.1016/j.jtbi.2024.111934. Epub 2024 Sep 4. PMID: 39241821.

Dhawale N, Mandre S, Venkadesan M. 2019 Dynamics and stability of running on rough terrains. R. Soc. open sci. 6: 181729. http://dx.doi.org/10.1098/rsos.181729

These are simpler or different models, but maybe at least a comment on why finding analytical or approximate solutions is a further challenge wrt what you are addressing in the paper would be useful. It feels to me this might also help your struggles in finding limit cycles numerically, as at least you can guess where parameters for stability are more easily.

Stability.

Your result in Figure 5 is not surprising, but there is something I am confused about. Depending on the controller, you should be able to cover all the parameter space with stable regions, or not? I think a comment on this early on would make your Figure 5 experiment a little more meaningful, otherwise I do not completely understand the use of it.

Feedback Policy.

Why as an evaluation method you used the root mean square error between the tracked path and the target gait only and did not include the RMSE also of the velocity curves?

Neural Network Controller.

I do not understand this section. I think you jumped from your controller to a feed forward neural network with a not too deep, arbitrary architecture, while there are many other options to compare your controller with other NN ones. It just seems unmotivated and there does not seem to me that there are strong reasons for which this was chosen.

The way the data is split into training and validation and testing is quite obscure. Of the 90% what percentage is training, what is validation? How was the split determined? Also the number of epochs seems arbitrary and in situation like this, where the loss function is very non convex, I would not be to confident that if you go from 200 epochs to 2000 epochs, the algorithm that seemed to converge to a specific gait does not end up visiting another gait.

I understand it is tempting to use a NN for this, but there are some cons to consider. First of all, whatever architecture you choose, there is not really a good motivation to support it. I know you mention biologically plausibility, but the network you are using is not more plausible than any others in practice and not even more plausible of more simple models. The fact that you save parameters in your architecture, well it’s because you chose that architecture and that policy, but, then, convincingly explaining that a randomly chosen NN architecture with smaller size saves parameters and matches your policy in terms of efficiency is cumbersome and this fact is quite a negative message about your policy.

The training-validation-test split is quite puzzling no matter what. I know cross-validation might be computationally expensive, but at least you might want to do a second split with the same percentages and similar other two splits with a different percentage in the split (start with one random seed for the first percentage split; then change the seed and repeat). The fact is that CV itself is not theoretically well understood in the iid case and here is so much more complicated that you risk that the true confidence bands of your estimates are enormous.

At the end, although you are doing your best to be rigorous, you might end up adding a section that damages the paper. It might sound extreme, but there are reasons that make me recommend removing this section (anything NN related) in full.

Some of your conclusions.

Isn’t it already known that open-loop stability is insufficient for gait adaptation? I mean it is already something rare without adaptation involved I would say. I think there needs to be a stronger or more precise conclusion on this, otherwise you might probably benefit from removing this paragraph.

When you add further parameters, you surely increase stability. You made a choice that works, but also this does not seem surprising to me. I think a stronger motivation of why your way is a better way to proceed is definitely needed. What I am trying to say in different words is that you are comparing no-control with control and of course this increases the stability region, basically no matter what type of control you use. Am I confused? If not, I think the motivation and the explanations of why your results are a step forward need to be stronger. And on top of this, there is the issue above of a randomly chosen NN with less parameters performing as efficiently as your policy.

Reviewer #2: This study investigates the controllability of a simple walking model with one or two simple controllers. It argues that while open-loop control cannot achieve human-like step length and step frequency, it is possible to design a system by adjusting the control inputs for each step. The study comprehensively investigates control performance using the model, and the theoretical development is carefully presented. Overall, the paper is well organized. However, while reading the paper, I encountered a few points that were unclear.

One of the main claims of this study is that open-loop control cannot control the entire range of human step length and step frequencies. I agree that this claim is assumed to be correct. The issue is whether this claim is fully supported by the results. I understand that the method used to demonstrate this is to numerically calculate the model's behavior to generate a limit cycle and then analyze the Poincare map around that limit cycle. This limit cycle seems to depend on the initial values, where the paper states that the initial values have been explored through trial and error (lines 152-153). My question is whether this trial-and-error exploration has any effect on the results. Personally, I think that it is probably not a problem in terms of the overall trend, but I have doubts about whether it is possible to make a statement as in line 469, "we showed that ... cannot fully control the ...". If necessary, I would like to see some discussion on whether there is any dependence on the initial values.

The second point concerns the evaluation method for the control system. In "Performance of the feedback policy" in the "Method" section, optimization is used to determine the response time. I found it difficult to understand how this was decided, especially the evaluation function and decision coefficient in lines 258-261, which seemed to appear suddenly. I felt that a more detailed explanation was needed for this part.

Minor comments:

* Line 166: It would be helpful to explain how the disturbance magnitude of 1e-7 was determined. If possible, it would be helpful to know how much influence this 1e-7 has on the motion.

* An explanation of the meaning of x in equation (4) would be desirable. I assume that this x is the same as the x in equation (3), but this was not clear when I first read the manuscript.

* Line 629, Reference 17: The year of publication is missing.

Reviewer #3: The authors investigate simple walking controllers which are sufficient for adapting to various step lengths and frequencies. Their methodology involved: generating a grid of limit-cycle gaits, designing feedback policies for these gaits, and testing two interpolation methods; linear and nonlinear. I find this work methodologically novel and interesting as well as rigorous. I do have some suggestions to further improve the rigor and clarity of the findings below:

One notable and surprising finding is the necessity of a third actuation, namely a second torque during the latter half of swing, to achieve control across the full gait pattern space. The paper concludes that a single hip spring and a single push-off are insufficient for such controllability. They show that adding a second hip spring with independent stiffness solves this problem and explains aspects of the swing phase hip muscle activation patterns. Especially because this is an interesting finding, it would be good to test or at least consider alternative explanations or controllers other than a second hip torque: have the authors tested a single spring and damper instead? Can they propose other controllers that may achieve similar stability?

The study also provides an interesting comparison between the two interpolation methods. The neural network based interpolation successfully approximates the control policy but is inferior to the simpler linear interpolation policy in the challenging region of high step lengths. In this region, the neural network either required more steps to complete a path or failed entirely, while the interpolation policy succeeded. This result, while interesting, could be discussed more explicitly and in detail in terms of its biological significance. The authors make an intriguing connection to the findings in Bertram and Ruina, stating “people report difficulty…” at these challenging combinations. Can the authors elaborate what is the nature of this difficulty and how does it relate to the neural network’s difficulties in interpolation?

The authors could discuss more prominently how their assumption of a deadbeat and linear controller might affect their findings and conclusions. What if the controller was instead optimized for some other cost function or was a nonlinear feedback control law?

Overall, the paper's writing could be more direct. I recommend simplifying complex sentences and using active voice to improve the clarity of the writing. The logical flow of the paragraphs can also be improved.

The methodology for evaluating the feedback policy's tracking performance could be more clearly articulated. While the paper mentions that the walker "starts from the initial gait" to track a path, it does not specify how this initial gait is chosen or if it is varied across different trials. The policy's performance could be dependent on the initial conditions, so clarifying this aspect would strengthen the analysis.

**Do you want your identity to be public for this peer review?** For information about this choice, including consent withdrawal, please see our Privacy Policy

Reviewer #1: No

Reviewer #2: No

Reviewer #3: No

---

## [Author Response · Author response to Decision Letter 1]

31 Oct 2025

PONE-D-25-37501: Controlling a simple model of bipedal walking to adapt to a wide range of target step lengths and step frequencies.

Dear Professor Martin,

We are pleased to submit the revised version of our manuscript entitled “Controlling a simple model of bipedal walking to adapt to a wide range of target step lengths and step frequencies.”

We sincerely thank you and the reviewers for the thoughtful and constructive feedback. We have carefully addressed all reviewer comments and implemented their suggestions throughout the manuscript. These revisions have improved the clarity, rigor, and overall presentation of the paper.

Please see, in the following, the point-by-point response to all reviewers’ comments and the edits made to the manuscript. We have also attached a revised version of the manuscript implementing all the changes reviewers suggested.

Thank you for considering our revised submission. We appreciate the opportunity to improve our work and look forward to your response.

Sincerely,

Sina Mehdizadeh

(on behalf of all co-authors)

Reviewer #1:

PONE-D-25-37501 Review

Thank you for the opportunity to review this interesting paper.

Here are some comments about the content of the manuscript.

Model. I feel that some analytical work on your model 1a,1b,1c can be done, instead of relying only on numerics. At the end, the angle swept during stance should be relatively easy to compute especially under some parameter regimes and this should be enough to determine, at least, an approximate apex return map for the model. See for example what is done in these papers:

Geyer H, Seyfarth A, Blickhan R. Spring-mass running: simple approximate solution and application to gait stability. J Theor Biol. 2005 Feb 7;232(3):315-28. doi: 10.1016/j.jtbi.2004.08.015. PMID: 15572057.

Selvitella AM, Foster KL. An approximate solution of the SLIP model under the regime of linear angular dynamics during stance and the stability of symmetric periodic running gaits. J Theor Biol. 2024 Dec 7;595:111934. doi: 10.1016/j.jtbi.2024.111934. Epub 2024 Sep 4. PMID: 39241821.

Dhawale N, Mandre S, Venkadesan M. 2019 Dynamics and stability of running on rough terrains. R. Soc. open sci. 6: 181729. http://dx.doi.org/10.1098/rsos.181729

These are simpler or different models, but maybe at least a comment on why finding analytical or approximate solutions is a further challenge wrt what you are addressing in the paper would be useful. It feels to me this might also help your struggles in finding limit cycles numerically, as at least you can guess where parameters for stability are more easily.

Authors’ answer:

We thank you for this insightful comment and fully agree that approximate analytical treatments—such as those developed for the SLIP model—offer valuable intuition and can serve as useful tools to guide numerical work. Kuo’s Idealized Simple Model (Kuo, 2001) is another example, where analytical predictions based on linearization and other simplifications provide important insights.

Similarly, our model could be linearized in the regime of small stance angles, which opens the possibility for deriving closed-form solutions under those assumptions. These analytical approximations could, in turn, be used to generate feasible initial guesses for subsequent numerical root-finding procedures. However, given the wide range of gait parameters we explored—including combinations with large stance angles—such approximations would not be valid or feasible across the entire parameter space.

To acknowledge and incorporate this perspective, we have added the following paragraph to the Methods section:

Line 162:

“While we used numerical approaches to find the limit-cycle gaits, approximate analytical treatments of locomotion models—such as those developed for the Simple Linear Inverted Pendulum model (Geyer et al., 2005; Selvitella & Foster, 2024; Dhawale et al., 2019) or Kuo’s Idealized Simple Model (Kuo, 2001)— could provide valuable intuition and can guide numerical analyses. In principle, our model could also be linearized in the regime of small stance angles, enabling approximate return maps that could generate initial guesses for numerical root-finding. However, because our study spans a wide gait space that includes large stance angles and strongly coupled swing–stance dynamics, such approximations would not remain valid across all conditions. For this reason, we relied on numerical methods to ensure comprehensive coverage of the full gait space.”

Stability. Your result in Figure 5 is not surprising, but there is something I am confused about. Depending on the controller, you should be able to cover all the parameter space with stable regions, or not? I think a comment on this early on would make your Figure 5 experiment a little more meaningful, otherwise I do not completely understand the use of it.

Authors’ answer:

To your question: “Depending on the controller, you should be able to cover all the parameter space with stable regions, or not?”—the answer depends primarily on the actuation scheme. With only push-off and a single hip spring, controllability breaks down along the gait space’s uncontrollable manifold, and no controller—regardless of its design—can stabilize those gaits. However, adding a second hip spring restores full controllability across the entire gait space. That said, controllability alone does not guarantee large or useful basins of attraction. One key result illustrated in Figure 5 is that the size of these basins is significantly increased at each grid point, which is essential for the success of our interpolation-based control policy. In the revised manuscript, we have clarified these distinctions and explicitly addressed this point in the new lines below. These additions also respond to related concerns raised by Reviewer 3.

Line 519:

“Our results on stability after control (both maximum tolerable perturbation and the tracking performance) are based on using a linear deadbeat controller. Alternative controller design approaches—such as linear quadratic regulation (LQR), nonlinear controller design, or optimization-based methods—could produce different outcomes. We chose the deadbeat approach because it enables the walker to eliminate disturbances in a single step and avoids the computational overhead associated with some of these other methods. Further, our preliminary experiments did not reveal significant performance differences between the deadbeat controller and an LQR-based controller. Importantly, our design approach—offline computation of controller gains combined with online interpolation—is independent of the specific controller type and can be extended to incorporate other control strategies.”

Feedback Policy. Why as an evaluation method you used the root mean square error between the tracked path and the target gait only and did not include the RMSE also of the velocity curves?

Authors’ answer:

We are not sure exactly you mean by “velocity” here. We can think of two types of velocity in this analysis: gait velocity, and derivatives of step length and step frequency at each step. Regarding the former (gait velocity), we should emphasize that there was no explicit target gait velocity to track. However, since the goal of the tracking at each step was to match both the target step length and target step frequency, this ensures tracking of gait velocity because gait velocity = step length X step frequency. Regarding the latter case (the time derivative of step length and step frequency): since the target gait is changing at each step, we wanted to allow the velocity to be whatever it needed to be to accurately track the path.

Neural Network Controller. I do not understand this section. I think you jumped from your controller to a feed forward neural network with a not too deep, arbitrary architecture, while there are many other options to compare your controller with other NN ones. It just seems unmotivated and there does not seem to me that there are strong reasons for which this was chosen.

The way the data is split into training and validation and testing is quite obscure. Of the 90% what percentage is training, what is validation? How was the split determined? Also the number of epochs seems arbitrary and in situation like this, where the loss function is very non convex, I would not be to confident that if you go from 200 epochs to 2000 epochs, the algorithm that seemed to converge to a specific gait does not end up visiting another gait.

I understand it is tempting to use a NN for this, but there are some cons to consider. First of all, whatever architecture you choose, there is not really a good motivation to support it. I know you mention biologically plausibility, but the network you are using is not more plausible than any others in practice and not even more plausible of more simple msodels. The fact that you save parameters in your architecture, well it’s because you chose that architecture and that policy, but, then, convincingly explaining that a randomly chosen NN architecture with smaller size saves parameters and matches your policy in terms of efficiency is cumbersome and this fact is quite a negative message about your policy.

The training-validation-test split is quite puzzling no matter what. I know cross-validation might be computationally expensive, but at least you might want to do a second split with the same percentages and similar other two splits with a different percentage in the split (start with one random seed for the first percentage split; then change the seed and repeat). The fact is that CV itself is not theoretically well understood in the iid case and here is so much more complicated that you risk that the true confidence bands of your estimates are enormous.

At the end, although you are doing your best to be rigorous, you might end up adding a section that damages the paper. It might sound extreme, but there are reasons that make me recommend removing this section (anything NN related) in full.

Authors Answer:

We agree with your concern and useful comments. Therefore, we have removed all the analyses, results, and conclusions related to the neural network policy to keep the coherency of the manuscript as you suggested.

Some of your conclusions. Isn’t it already known that open-loop stability is insufficient for gait adaptation? I mean it is already something rare without adaptation involved I would say. I think there needs to be a stronger or more precise conclusion on this, otherwise you might probably benefit from removing this paragraph.

Authors Answer:

You are correct—open-loop stability is not sufficient for gait adaptation, which is precisely why we designed and implemented the feedback controller. However, it was still necessary to document the performance of open-loop stability to then quantify how much adaptation performance increased by adding active control. To do so, we conducted two analyses: one evaluating the maximum tolerable perturbation and another assessing the response time to converge to target gaits.

Although we presented the results of these analyses in the Results section for completeness, we did not emphasize them as primary conclusions in the Discussion section for the very reason you noted. In this regard, we believe our interpretation and your perspective are well aligned with respect to the main contributions of our work.

When you add further parameters, you surely increase stability. You made a choice that works, but also this does not seem surprising to me. I think a stronger motivation of why your way is a better way to proceed is definitely needed. What I am trying to say in different words is that you are comparing no-control with control and of course this increases the stability region, basically no matter what type of control you use. Am I confused? If not, I think the motivation and the explanations of why your results are a step forward need to be stronger. And on top of this, there is the issue above of a randomly chosen NN with less parameters performing as efficiently as your policy.

Authors Answer:

Thank you for your thoughtful observation and the opportunity to clarify the motivation and novelty of our approach. We do not believe it is self-evident that simply adding arbitrary control will enhance stability relative to no control; the effectiveness of control depends critically on how it is designed. Moreover, our findings demonstrate that even an optimally designed controller may fail to stabilize the system in certain regions of the gait space if actuation is limited to once-per-step adjustments of push-off and a single hip spring.

Importantly, identifying that stability regions increase with appropriate control is not sufficient to address the more fundamental question we pose: Can this walker adapt at human-like time scales across the entire gait space? Our work directly addresses this question and provides a systematic framework to answer it.

Overall, our work goes beyond this basic no-control/control comparison in at least two key ways:

We show that with only a single hip spring and push-off impulse—the conventional actuation used for steady-state walking—it is mathematically impossible to control all gaits in the gait space. Specifically, we demonstrate a zero-boundary where stance angle cannot be influenced, and the controllability matrix becomes rank deficient. This means that no linear feedback scheme—irrespective of gain choice—can stabilize those gaits under the single-spring model. Adding a second independent hip spring is not simply “more parameters,” but the minimal structural change needed to make the system fully controllable across the entire range of human-like gaits. This is a qualitative improvement, not just a quantitative one.

Our focus is on whether once-per-step controllers can enable adaptation between diverse gaits within human-like timescales. We show that open-loop stable gaits converge only slowly (hundreds of steps), whereas our deadbeat once-per-step feedback controllers achieve convergence in 2–5 steps for most gaits, closely matching empirical human adaptation studies. Thus, the step forward is not simply expanding the “stability region,” but designing a feedback strategy that enables rapid, robust adaptation across the full gait space using once-per-step control.

We have elaborated these points in both the abstract and first paragraph of the discussion:

Line 14:

“We tested whether the same control principles that support steady-state walking are sufficient for robust, rapid gait adaptation over a wide range of step lengths and frequencies. We begin by demonstrating that periodic gaits exist at combinations of step frequency and step length that span the full range of gaits achievable by humans. However, their open-loop stability is not enough to rapidly transition to target gaits. Next, we show that actuating with only one push-off and one hip spring of fixed stiffness cannot fully control the walker in the entire gait space. We solve this by adding a second hip spring with an independent stiffness to actuate the second half of the swing phase. This allowed us to design local feedback controllers that provided rapid convergence to target gaits by making once-per-step adjustments to control inputs. To adapt to a range of target gaits that vary over time, we interpolated between local controllers. This policy performs well, accurately tracking rapidly varying combinations of target step length and step frequency with human-like response times.”

Line 447:

“We demonstrated that the same method of actuation and control of steady-state walking are insufficient for robust and rapid gait adaptation over a wide range of step lengths and frequencies. Specifically, we showed that relying on only one push-off and one hip spring of fixed stiffness cannot fully control the walker in the entire gait space. Thus, we added a third actuation , allowing the controller to adjust the hip spring stiffness midway through the swing phase. This allowed us to design feedback controllers that stabilized periodic gaits by making once-per-step adjustments to push-off and hip spring stiffnesses providing convergence to target gaits within human-like time scales. Finally, we extended the controller

---

## [Decision Letter · Decision Letter 1]

25 Nov 2025

Controlling a simple model of bipedal walking to adapt to a wide range of target step lengths and step frequencies

PONE-D-25-37501R1

Dear Dr. Mehdizadeh,

We’re pleased to inform you that your manuscript has been judged scientifically suitable for publication and will be formally accepted for publication once it meets all outstanding technical requirements.

Kind regards,

Anne E. Martin

Academic Editor

PLOS ONE

Additional Editor Comments (optional):

Reviewers' comments:

Reviewer's Responses to Questions

**Comments to the Author**

Reviewer #1: All comments have been addressed

Reviewer #2: All comments have been addressed

Reviewer #3: All comments have been addressed

2. Is the manuscript technically sound, and do the data support the conclusions?

Reviewer #1: Yes

Reviewer #2: Yes

Reviewer #3: Yes

3. Has the statistical analysis been performed appropriately and rigorously?

Reviewer #1: Yes

Reviewer #2: Yes

Reviewer #3: Yes

4. Have the authors made all data underlying the findings in their manuscript fully available?

Reviewer #1: (No Response)

Reviewer #2: Yes

Reviewer #3: Yes

5. Is the manuscript presented in an intelligible fashion and written in standard English?

Reviewer #1: Yes

Reviewer #2: Yes

Reviewer #3: Yes

Reviewer #1: The authors have addressed all the main concerns I had. In particular, I appreciate the explanations about the role of the control used and the decision of removing the neural network component of the work. Thank you for the opportunity to review this manuscript.

Reviewer #2: I am grateful to the authors for their sincere response to the items I commented on. I have no further comments to add.

Reviewer #3: The authors have adequately addressed my comments. I still believe this work is interesting and insightful. look forward to seeing this work in print.

**Do you want your identity to be public for this peer review?** For information about this choice, including consent withdrawal, please see our Privacy Policy

Reviewer #1: No

Reviewer #2: No

Reviewer #3: No

---

## [Editor Report · Acceptance letter]

PONE-D-25-37501R1

PLOS One

Dear Dr. Mehdizadeh,

I'm pleased to inform you that your manuscript has been deemed suitable for publication in PLOS One. Congratulations! Your manuscript is now being handed over to our production team.

Kind regards,

on behalf of

Dr. Anne E. Martin

Academic Editor

PLOS One